# Gating of neural error signals during motor learning

Rhea R Kimpo[†], Jacob M Rinaldi[†], Christina K Kim[†], Hannah L Payne[†], Jennifer L Raymond*

Department of Neurobiology, Stanford University, Stanford, United States

**Abstract** Cerebellar climbing fiber activity encodes performance errors during many motor learning tasks, but the role of these error signals in learning has been controversial. We compared two motor learning paradigms that elicited equally robust putative error signals in the same climbing fibers: learned increases and decreases in the gain of the vestibulo-ocular reflex (VOR). During VOR-increase training, climbing fiber activity on one trial predicted changes in cerebellar output on the next trial, and optogenetic activation of climbing fibers to mimic their encoding of performance errors was sufficient to implant a motor memory. In contrast, during VOR-decrease training, there was no trial-by-trial correlation between climbing fiber activity and changes in cerebellar output, and climbing fiber activation did not induce VOR-decrease learning. Our data suggest that the ability of climbing fibers to induce plasticity can be dynamically gated in vivo, even under conditions where climbing fibers are robustly activated by performance errors.

## Introduction

### Overview

The cerebellum is thought to implement a supervised learning algorithm, with the climbing fiber input to the cerebellum providing error signals that induce learning. According to this model, performance errors activate neurons in the inferior olive and their climbing fiber axons, which in turn trigger the induction of plasticity in the cerebellar cortex to produce adaptive changes in behavior (*Marr, 1969*; *Albus, 1971*). Previous work has shown that this process is regulated by a feedback loop from the cerebellar cortex to the inferior olive (*Andersson and Armstrong, 1987*; *Hesslow and Ivarsson, 1994*; *Medina et al., 2002*; *Rasmussen et al., 2008*; *Yang and Lisberger, 2013*; reviewed in *Apps, 1999*; *Gibson et al., 2002*). In this study, we describe a second level of regulation in animals undergoing learning: even when climbing fibers are robustly activated by performance errors, the ability of that climbing fiber activity to trigger plasticity can be regulated by the state of the cerebellar circuit. Thus, the cerebellum is not a slave to its climbing fiber 'teachers', but rather plays an active role in determining whether it will adapt in response to the error signals it receives from the climbing fibers.

### Climbing fiber activity as the trigger of cerebellar learning: conflicting evidence

The question of whether error signals carried by climbing fibers are the trigger for learning has been controversial, since the evidence from different studies has been inconsistent. Climbing fibers are activated by performance errors during a wide range of motor learning tasks (for a review, see *Ito, 2001*), and several lines of evidence suggest that this provides a potent trigger for cerebellar plasticity. A single spike in a climbing fiber reliably triggers a complex spike in its Purkinje cell targets (*Eccles et al., 1966*). Calcium transients associated with complex spikes can induce synaptic plasticity in the cerebellar cortex in vitro (for reviews, see *Hansel et al., 2001*; *Ito, 2001*). In vivo, climbing fiber

*For correspondence:
jenr@stanford.edu

[†]These authors contributed equally to this work

Competing interests: The authors declare that no competing interests exist.

**eLife digest** The cerebellum (or 'little brain') is located underneath the cerebral hemispheres. Despite comprising around 10% of the brain's volume, the cerebellum contains roughly half of the brain's neurons. Many of the functions of the cerebellum are related to the control and fine-tuning of movement, and people whose cerebellum has been damaged have problems with balance and coordination, and with learning new motor skills.

One of the roles of the cerebellum is to control a reflex known as the vestibulo-ocular reflex, which enables us to keep our gaze fixed on an object as we turn our heads. The cerebellum relays information about head movements to the muscles that control the eyes, instructing the eyes to move in the opposite direction to the head. This keeps the image of the object we are looking at stable on the retina.

The vestibulo-ocular reflex is controlled by a circuit that includes Purkinje cells (which are the main output cells of the cerebellum) and climbing fibres (which originate in the brainstem). Any failure of the vestibulo-ocular reflex to fully compensate for head movements generates an error signal that activates the climbing fibres. These in turn modify the output of Purkinje cells, leading ultimately to adjustments in eye movements.

However, Kimpo et al. have now obtained evidence that Purkinje cells can modulate their response to the instructions they receive from climbing fibres. Monkeys sat in a rotating chair while a visual object they were trained to track with their eyes was moved to induce errors in the vestibulo-ocular reflex. When the object was moved so that a bigger reflexive eye movement was required to stabilize the image, the activation of the climbing fibres in response to the error led to a change in the response of the Purkinje cells, as expected. However, when a smaller reflexive eye movement was needed, the error-driven responses of the climbing fibres did not alter the responses of Purkinje cells. Similar results were obtained using pulses of light to artificially activate climbing fibres and thus simulate error signals.

The work of Kimpo et al. indicates that the cerebellum does not blindly follow the instructions it receives from the brainstem, but can instead modulate its responses to incoming information about performance errors. Further work is now required to identify factors that influence the responsiveness of the cerebellum: such information could ultimately be used to improve learning of motor skills and recovery from injury.

activation can replace the unconditioned stimulus used to induce a form of cerebellum-dependent classical conditioning (*Mauk et al., 1986*; *Steinmetz et al., 1989*; *Jirenhed et al., 2007*; *Rasmussen et al., 2013*) or the sensory feedback used to induce oculomotor learning (*Nguyen-Vu et al., 2013*). Moreover, during a smooth pursuit oculomotor learning task, there is a tight, trial-by-trial correlation between the occurrence of individual spikes in a climbing fiber and changes in cerebellar output on the subsequent trial, suggesting that climbing fiber spikes provide a potent and reliable trigger for plasticity (*Medina and Lisberger, 2008*; *Yang and Lisberger, 2010*, *2013*).

Despite the considerable evidence that climbing fiber activity provides error signals controlling motor learning, several studies have called this idea into question. Recordings during some cerebellum-dependent tasks have revealed a dissociation between the encoding of errors and the induction of learning over the course of a training session (*Catz et al., 2005*; *Ke et al., 2009*; *Kitazawa et al., 1998*; *Ojakangas and Ebner, 1992*). In addition, perturbation of the most extensively studied form of climbing fiber-triggered plasticity, long-term depression of the parallel fiber-to-Purkinje cell synapses (pf-Pk LTD), impairs certain learning tasks while leaving other cerebellum-dependent learning tasks intact (*Aiba et al., 1994*; *Boyden et al., 2006*; *De Zeeuw et al., 1998*; *Feil et al., 2003*; *Hansel et al., 2006*; *Katoh et al., 2000*; *Katoh et al, 2005*; *Kim and Thompson, 1997*; *Kishimoto et al., 2001*; *Kishimoto et al., 2001*; *Koekkoek et al., 2003*; *Schonewille et al., 2011*; *Shibuki et al., 1996*; *Shutoh et al., 2002*, *2003*; *van Alphen and De Zeeuw, 2002*; *Welsh et al., 2005*; *Yanagihara and Kondo, 1996*). Thus, after decades of research, the conflicting evidence has left unresolved the question of whether error signals in the climbing fibers are the driver of motor learning.

A potential reconciliation of the seemingly inconsistent results in the literature is suggested by recent studies in reduced preparations. Climbing fiber spikes were originally thought to convey a

reliable, invariant, all-or-none signal for plasticity. However, more recently, it has been shown that a Purkinje cell's response to its climbing fiber input can be graded, which, in turn, can regulate the efficacy of climbing fiber stimulation to induce plasticity in vitro and in decerebrate preparations (*Weber et al., 2003*; *Carey and Regehr, 2009*; *Maruta et al, 2007*; *Mathy et al., 2009*; *Rasmussen et al., 2013*). However, to date, there has been no evidence about whether the ability of the error signals carried by climbing fibers to induce learning is also regulated in awake, behaving animals. When climbing fibers are activated by performance errors, does this reliably trigger plasticity, or can the impact of these error signals in the climbing fibers be gated? To address this question, we compared two closely related oculomotor learning paradigms that rely on the same cerebellar microcircuit: learned increases and decreases in the gain of the vestibulo-ocular reflex (VOR; *Figure 1*, *Figure 2A,B*; for a review, see *Boyden et al., 2004*).

## Error signals in the climbing fibers during vestibulo-ocular reflex (VOR) learning

The VOR functions to stabilize visual images on the retina by using vestibular signals to elicit eye movements that compensate for head movements. To successfully stabilize images, the gain of the VOR (amplitude of the eye movement response to a vestibular input) needs to be well calibrated. This calibration occurs through a form of motor learning that depends on the cerebellar floccular complex (flocculus and ventral paraflocculus) (*Robinson, 1976*; *Ito et al., 1982*; *Lisberger et al, 1984*; *Nagao, 1983*; *McElligott et al., 1998*; *Rambold et al., 2002*). Climbing fibers in this part of the cerebellum are robustly activated by performance errors during both VOR-increase and VOR-decrease learning: if the eye movements driven by the VOR are too small or too big, that performance error results in image motion on the retina (retinal slip), which is encoded by the floccular climbing fibers (*Ghelarducci et al., 1975*; *Graf et al., 1988*; *Simpson and Alley, 1974*; *Stone and Lisberger, 1990*; *Figure 2C–F* and *Figure 2—figure supplement 1C–F*). The same climbing fibers encode performance errors during both VOR-increase and VOR-decrease learning.

Previous efforts to determine whether these error signals in the climbing fibers are what drive VOR learning have been inconclusive. Over the course of both VOR-increase and VOR-decrease learning, the Purkinje cell simple spike output during the VOR is altered in a manner consistent with the induction of LTD in parallel fiber inputs that were coactive with the climbing fibers during training

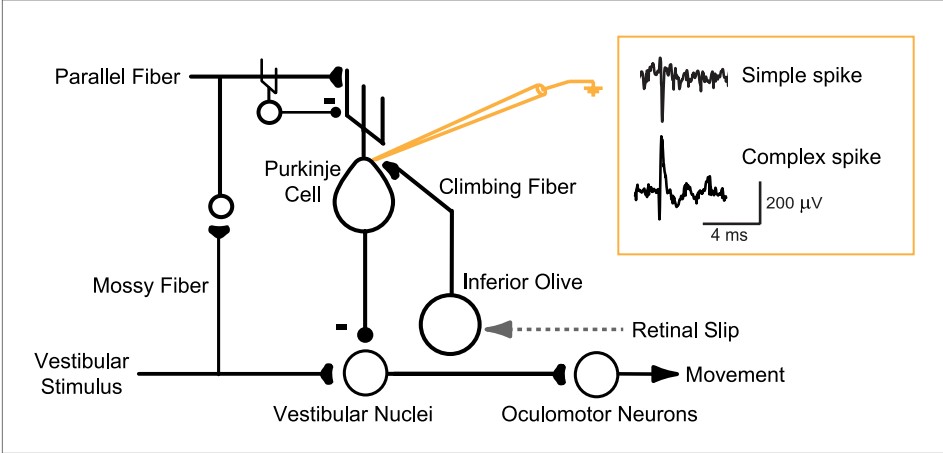

**Figure 1**. Circuit for VOR motor learning. Vestibular stimuli (head movements) drive eye movement responses through VOR interneurons in the vestibular nuclei. Vestibular signals are also conveyed, via parallel fibers and interneurons, to Purkinje cells in the cerebellar floccular complex. Purkinje cells also receive input from climbing fibers that respond to retinal slip, which indicates a performance error--a failure of eye movements to stabilize a visual image on the retina. Climbing fiber activity is hypothesized to drive plasticity in the other inputs to Purkinje cells. Changes in Purkinje cell output can influence eye movements through their inhibitory effect on VOR interneurons in the vestibular nuclei. An extracellular recording from a Purkinje cell can detect complex spikes, which reflect spikes in its single climbing fiber input with a one-to-one correspondence, and simple spikes (71 ± 9 sp/s, mean ± SEM), which greatly outnumber complex spikes (0.98 ± 0.19 sp/s) and thus are the major output from the Purkinje cells.

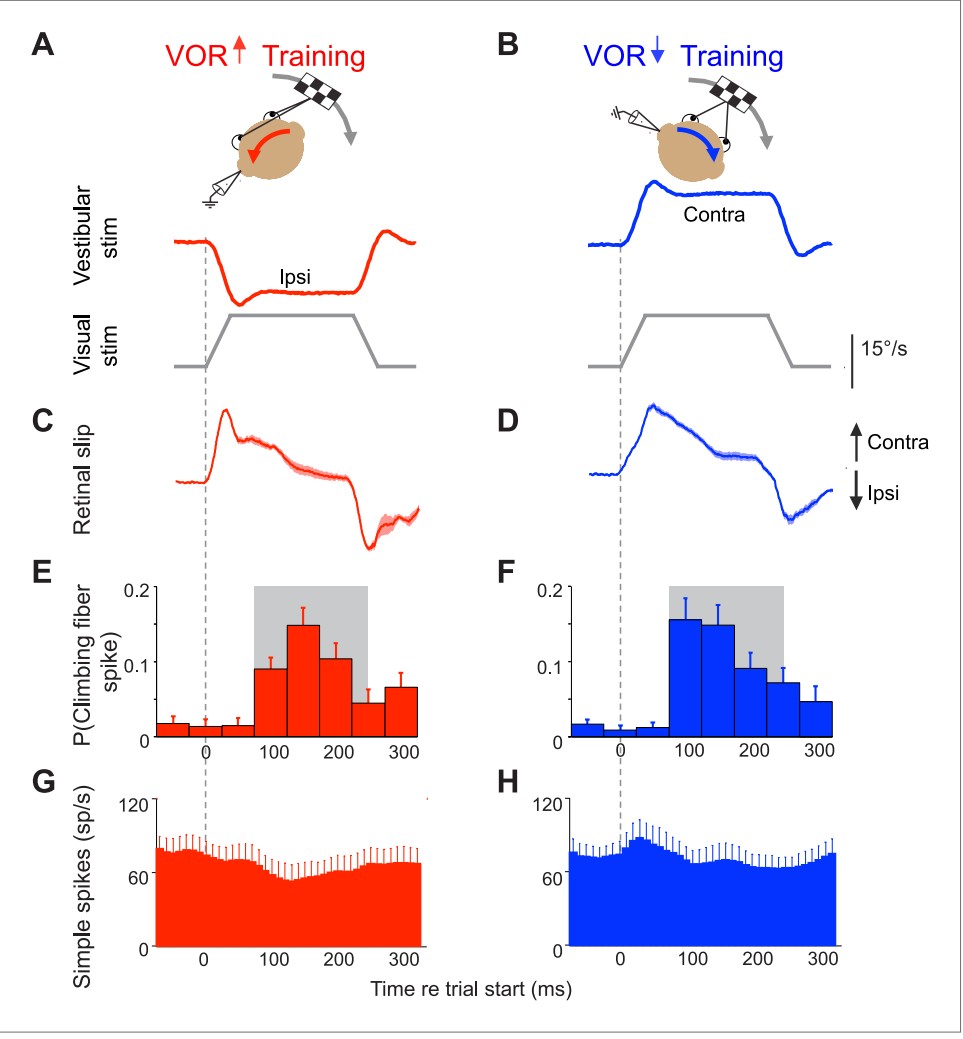

**Figure 2**. Climbing fibers encode errors during VOR motor learning. (**A** and **B**) VOR learning is induced by pairing a vestibular stimulus with a moving visual stimulus. During VOR-increase training (**A**), the visual stimulus (grey trace) is paired with a vestibular stimulus in the opposite direction (red). During VOR-decrease training (**B**), the visual stimulus is paired with a vestibular stimulus moving in the same direction (blue). In panels **A**–**D**, upward and downward deflections indicate contraversive and ipsiversive motion, respectively, as defined relative to the side of the brain on which the neural recordings in panels **E**–**H** were made: if the climbing fiber was recorded from the left cerebellar flocculus, then leftward head rotations and leftward image motion on the retina were defined as ipsiversive. (**C** and **D**) During the visual-vestibular training stimuli, there is retinal slip, reflecting the failure of the eye movements to stabilize the visual stimulus, which is a performance error. Retinal slip is plotted in °/s, same scale as **A**, **B**. (**E** and **F**) Responses of the climbing fibers during vestibular stimuli in the direction accompanied by *contraversive* retinal slip (the 'on' direction), which drives an increase in firing in the climbing fibers. During VOR-increase training (**E**), the probability of a climbing fiber spike was elevated in a window 75–250 ms (grey box) after the onset of each *ipsiversive* vestibular stimulus (i.e., climbing fiber activity in the left flocculus increased during head rotation to the left). During VOR-decrease training (**F**), the same climbing fibers increased their firing probability 75–250 ms after the onset of a *contraversive* vestibular stimulus. All climbing fibers that responded during one training paradigm also responded during the other paradigm, and with a similar response amplitude (t(9) = 1.77, p=0.11 paired *t*-test). For both training paradigms, climbing fiber activity was suppressed during interleaved vestibular stimuli in the opposite direction from those shown here (***Figure 2—figure supplement 1E,F***). (**G** and **H**) The corresponding Purkinje cell simple spike firing rate.

The following figure supplements are available for figure 2:

**Figure supplement 1**. Climbing fiber responses during stimuli in the 'off' direction.

(*Dufossé et al., 1978*; *Hirata and Highstein, 2001*; *Lisberger et al., 1994*; *Miles et al., 1980*; *Nagao, 1989*; *Watanabe, 1984*, *1985*). Also, direct, optogenetic activation of the climbing fibers can induce VOR-increase learning, when paired with a vestibular stimulus (*Nguyen-Vu et al., 2013*). On the other hand, a previous attempt to induce VOR-decrease learning with climbing fiber stimulation was unsuccessful (*Nguyen-Vu et al., 2013*). Moreover, studies of multiple lines of mice deficient in pf-Pk LTD have not consistently found impairments of VOR learning, and the reported impairments are more pronounced for VOR-increase than VOR-decrease learning (*Aiba et al., 1994*; *Boyden et al., 2006*; *De Zeeuw et al., 1998*; *Feil et al., 2003*; *Hansel et al., 2006*; *Kim and Thompson, 1997*; *Koekkoek et al., 2003*; *Schonewille et al., 2011*; *van Alphen and De Zeeuw, 2002*). These results raised the possibility that the efficacy of the error signals carried by climbing fibers to induce plasticity is reduced during VOR-decrease training. In this study, we provide convergent evidence for this possibility from recording and stimulation experiments. The regulation of climbing fiber efficacy for inducing plasticity represents a new component of the learning algorithm implemented by the cerebellum.

## Results

### Trial-by-trial relationship between climbing fiber activity and plasticity

We recorded from Purkinje cells in the floccular complex of two adult rhesus monkeys during VOR-increase and VOR-decrease training. An extracellular recording from a Purkinje cell provides simultaneous access to two distinct physiological signals: complex spikes, which provide a one-to-one readout of spikes in the single climbing fiber innervating the Purkinje cell (*Eccles et al., 1966*); and simple spikes, which reflect the impact of all excitatory and inhibitory inputs as well as the intrinsic excitability of the Purkinje cell (*Figure 1*).

The climbing fiber input to Purkinje cells in the flocculus encodes the retinal slip that drives VOR learning (*Simpson and Alley, 1974*; *Ghelarducci et al., 1975*; *Graf et al., 1988*; *Stone and Lisberger, 1990*). Retinal slip can induce an adaptive increase or decrease in the gain of the VOR, depending on its direction relative to the vestibular stimulus (*Boyden et al., 2004*). To induce a learned increase in VOR gain, visual image motion is paired with a vestibular stimulus in the opposite direction (*Figure 2A*); to induce a learned decrease in VOR gain, visual image motion is paired with a vestibular stimulus in the same direction (*Figure 2B*; *Collewijn and Grootendorst, 1979*; *Vercher and Gauthier, 1990–1991*; *Pastor et al., 1994*; *Raymond and Lisberger, 1996*). Most climbing fibers in the flocculus increase their firing in response to contraversive retinal slip (image motion away from the side on which the cell is recorded) (*Simpson and Alley, 1974*; *Raymond and Lisberger, 1998*; *Stone and Lisberger, 1990*; *Figure 2C–F*). During VOR-increase training, the contraversive retinal slip that activates climbing fibers occurs during *ipsiversive* vestibular stimuli (*Figure 2A,C,E* and *Figure 2—figure supplement 1A,C,E*), whereas during VOR-decrease training, the contraversive retinal slip and associated increase in climbing fiber activity occur during *contraversive* vestibular stimuli (*Figure 2B,D,F* and *Figure 2—figure supplement 1B,D,F*). The same climbing fibers encode retinal slip during both training paradigms, and the amplitude of the response is the same (*Figure 2E,F*). What distinguishes the two paradigms, and carries information about the required direction of learning, is whether the increased climbing fiber activity occurs during vestibular stimuli in one direction vs the other.

During each learning paradigm, the climbing fiber activity coincides with vestibular stimuli in the appropriate direction to potentially induce adaptive changes in the eye movements. Climbing fiber-induced plasticity can reduce Purkinje cell simple spike output, either through long-term depression (LTD) at the parallel fiber-Purkinje cell synapses (*Ito et al., 1982*) or through potentiation of inhibitory inputs to the Purkinje cell (*Jörntell and Ekerot, 2003*; *Kano et al., 1992*; *Tanaka et al., 2013*). The error signals carried by climbing fibers during VOR-increase and VOR-decrease training should trigger reductions in Purkinje cell output during ipsiversive and contraversive vestibular stimuli, respectively. A reduction in Purkinje cell firing during *ipsiversive* vestibular stimuli would cause VOR interneurons in the vestibular nuclei to receive inhibition from Purkinje cells that is more out-of-phase with the excitatory input they receive from vestibular afferents (*Figure 1*), leading to larger responses in the vestibular nuclei and hence an increase in VOR gain (*Ito, 1982*). In contrast, a climbing fiber-triggered reduction in Purkinje cell firing during *contraversive* vestibular stimuli would cause the inhibition from Purkinje cells to be more in-phase with the excitatory vestibular afferents, resulting in smaller responses of vestibular nuclei neurons and a decrease in VOR gain. Such changes in Purkinje cell responses have been reported by several laboratories after VOR learning (*Dufossé et al., 1978*; *Hirata and Highstein,*

2001; *Lisberger et al., 1994*; *Miles et al., 1980*; *Nagao, 1989*; *Watanabe, 1984*, *1985*). However, the question of whether the error signals in the climbing fibers are what triggers these changes has been controversial (*Ke et al., 2009*; *Nguyen-Vu et al., 2013*; reviewed in *Ito, 1982*; *Lisberger, 1988*).

One approach we used to address this question was a trial-by-trial analysis of the correlation between activity in the climbing fibers and plasticity of the Purkinje cell responses during training. The activity of an individual climbing fiber encodes retinal slip in a probabilistic manner (*Figure 2E,F*). Climbing fibers fire at a low rate, and even when retinal slip is present, an individual climbing fiber does not fire on every trial. We harnessed this natural variance to assess whether the activation of the climbing fibers during training is what drives plasticity in the VOR circuit. If a spike in the climbing fiber on one trial induces plasticity in the inputs to its Purkinje cell target, this might be detected as a change in the Purkinje cell's simple spike response on the next trial, as previously reported during smooth pursuit eye movement learning (*Medina and Lisberger, 2008*; *Yang and Lisberger, 2010*, *2013*). Therefore, we assessed the extent to which the presence or absence of a spike in its climbing fiber input on one trial predicted a change in a Purkinje cell's simple spike response on the subsequent trial during VOR learning.

We identified pairs of consecutive trials in which there was a spike in the climbing fiber on the first trial of the pair, but not the second trial (CF–No CF pairs), and calculated the trial-to-trial change in the simple spike response (*Figure 3A*). Likewise, we identified pairs of consecutive trials in which there was no spike in the climbing fiber on either trial (No CF–No CF pairs), and calculated the trial-to-trial change in the simple spike response. During VOR-increase training, there was a tight, trial-by-trial correlation between the activity of the climbing fiber input to a given Purkinje cell and the induction of changes in its simple-spike output (*Figure 3B*). If there was a spike in the climbing fiber input on the first trial of a pair, there was a reduction of about 8 spikes/s in the firing rate of the Purkinje cell on the subsequent trial (*Figure 3B,C*; CF–No CF, red). On trials with no climbing fiber spike, there was no reduction in Purkinje cell firing rate on the subsequent trial (*Figure 3B,C*; No CF–No CF, black). The sensory error (retinal slip speed) was indistinguishable on trials with a climbing fiber spike vs without a climbing fiber spike, and therefore cannot account for the difference observed on the trial after the climbing fiber spike (*Figure 3—figure supplement 1*).

The tight, trial-by-trial correlation between the occurrence of a climbing fiber spike on one trial, and the change in Purkinje cell simple spike output on the next trial suggests that the activation of the climbing fibers is what is triggering the changes in Purkinje cell output during VOR-increase training. In striking contrast, in the very same set of cells, a trial-by-trial analysis during VOR-decrease learning revealed no correlation between climbing fiber activity and plasticity of Purkinje cell responses (*Figure 3D,E*; CF–No CF, blue). Unlike VOR-increase training and all previous trial-by-trial analyses of smooth pursuit learning (*Medina and Lisberger, 2008*; *Yang and Lisberger, 2010*, *2013*), there was no reduction in Purkinje cell firing on the trial after a climbing fiber spike during VOR-decrease training (*Figure 3D,E*; CF-No CF, blue).

The Purkinje cells did undergo gradual, adaptive changes in their responses over the course of the full VOR-decrease training session, which could be detected during the same ~90-s VOR-decrease training sessions used for the trial-by-trial analysis (*Figure 4*). Across trials of VOR-decrease training, there was a progressively lower Purkinje cell firing rate during contraversive vestibular stimuli, similar to what has been reported previously using much longer training periods (*Dufossé et al., 1978*; *Hirata and Highstein, 2001*; *Lisberger et al., 1994*; *Miles, Braitman, et al., 1980*; *Nagao, 1989*; *Watanabe, 1984*, *1985*). This observation, along with the observation that lesions of the flocular complex disrupt both VOR-increase and VOR-decrease learning (*Ito et al., 1982*; *Koekkoek et al., 1997*; *Lisberger et al., 1984*; *Nagao, 1983*; *Rambold and Churchland, 2002*), indicates that the Purkinje cells in our sample participate in VOR-decrease learning as well as VOR-increase learning. However, there was no trial-by-trial correlation between the climbing fiber activity and the plasticity of the Purkinje cell responses to suggest a causal relationship during VOR-decrease training, as there was during VOR-increase learning. Thus, although the performance errors elicited equally robust responses in the climbing fibers during the two learning paradigms, those responses seem to have a different impact in terms of their ability to induce plasticity.

What could be preventing the error signals in the climbing fibers from inducing adaptive changes during VOR-decrease training? One trivial possibility would be that during contraversive vestibular stimuli there are simply not enough synapses active simultaneously with the climbing fibers to serve as a substrate for associative plasticity. This seems unlikely because there is robust activation of mossy

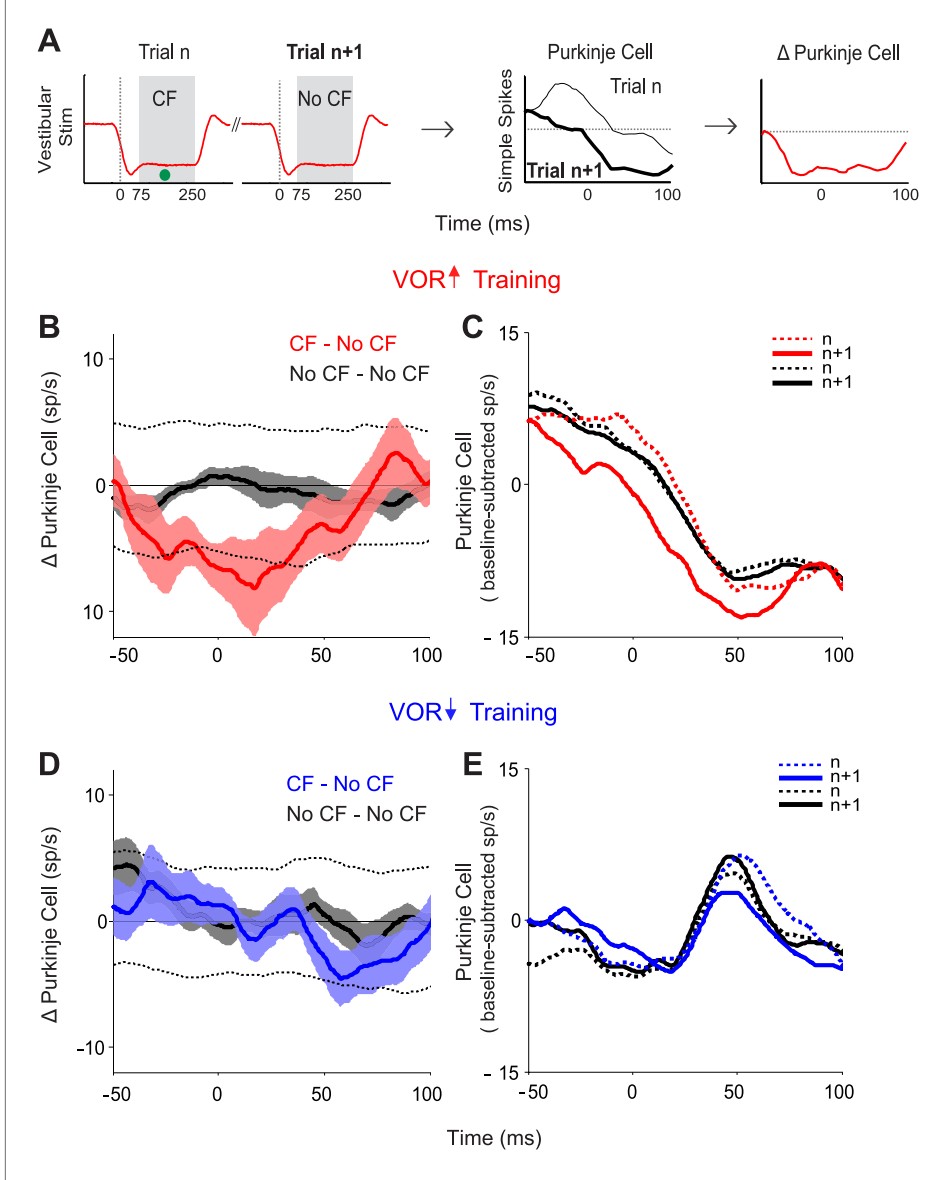

**Figure 3**. Trial-by-trial effects of climbing fiber activity on Purkinje cell output. (**A**) Schematic illustrating the analysis. Isolated Purkinje cells were recorded during the VOR training paradigms illustrated in *Figure 2*. Pairs of consecutive trials were identified in which a complex spike, indicative of a spike in the climbing fiber input to the cell (•, CF), occurred 75–250 ms after stimulus onset in the first trial of the pair, but not in the subsequent trial (left, CF–No CF pair). The Purkinje cell's simple spike firing rate during the first trial was subtracted from the second trial to calculate the change in Purkinje cell response on the trial after the climbing fiber spike (right). This analysis was performed for stimuli in the 'on' direction for the climbing fiber: ipsiversive vestibular stimuli during VOR-increase training, and contraversive vestibular stimuli during VOR-decrease training. (**B**) Trial-to-trial changes in Purkinje cell responses during VOR-increase training. If there was a spike in the climbing fiber on the first trial of a pair, there was a decrease in Purkinje cell firing on the subsequent trial (CF–No CF pairs, red). If there was no spike in the climbing fiber on the first trial, there was no detectable change in Purkinje cell firing on the subsequent trial (No CF–No CF pairs, black). Changes were considered significant if they lay outside the 95% confidence interval from a bootstrap distribution of the data (shown for CF–No CF data, dashed black lines, see 'Materials and methods'). There were no significant changes in the No CF–No CF trial pairs based on their confidence interval. (**C**) The Purkinje cell firing rate (baseline subtracted) during the first (dashed lines) and second trials (solid lines) of CF–No CF (red) and No CF–No CF pairs (black) during VOR-increase training. (**D**) Trial-to-trial changes in Purkinje cell responses during VOR-decrease training. There was no reduction of Purkinje cell firing on the trial after a climbing fiber spike (CF–No CF, blue, dashed black lines are 95% confidence intervals from the bootstrap

*Figure 3. Continued on next page*

*Figure 3. Continued*

distribution). (**E**) The Purkinje cell firing rate (baseline subtracted) during the first (dashed lines) and second trials (solid lines) of CF – No CF (blue) and No CF – No CF pairs (black) during VOR-decrease training.

The following figure supplements are available for figure 3:

**Figure supplement 1**. Similar retinal slip in trials with vs without a climbing fiber spike.

---

fiber inputs to the cerebellar flocculus during both contraversive and ipsiversive vestibular stimuli (***Noda, 1986***), and a substantial fraction of Purkinje cells increase their firing in response to contraversive vestibular stimuli (***Dufossé et al., 1978***; ***Lisberger and Fuchs, 1978***; ***Miles et al., 1980***; ***Watanabe, 1984***; ***Nagao, 1989***; ***Pastor et al, 1997***; ***Lisberger et al., 1994***; ***Raymond and Lisberger, 1997***; ***Hirata and Highstein, 2001***; ***Blazquez et al., 2003***; ***Ke et al., 2009***). Thus, there should be cerebellar synapses coactive with the climbing fibers during both training paradigms, and therefore some other aspect of the state of the cerebellar circuit is more likely to be regulating the ability of the error signals carried by climbing fibers to induce plasticity.

Recent studies have highlighted the duration of the complex spike as a factor that may regulate the induction of plasticity by the climbing fibers. The complex spike consists of a large initial spike followed by a variable number of spikelets. When the number of spikelets, and hence the complex spike duration, is manipulated in vitro or in decerebrate animals (***Carey and Regehr, 2009***; ***Mathy et al., 2009***; ***Rasmussen et al., 2013***), the probability of climbing fiber-induced plasticity can be altered, with shorter complex spike durations associated with a lower probability of climbing fiber-induced plasticity. We evaluated whether shorter complex spike durations could explain the lack of a trial-to-trial effect of climbing fiber activity during VOR-decrease training, by analyzing the waveform of complex spikes during each training paradigm (***Figure 2E,F***), in the same analysis windows used for the trial-by-trial analysis. Surprisingly, we found that complex spike waveforms during VOR-decrease training had more spikelets and were of longer duration than those during VOR-increase training (***Figure 5A***). Thus, the reduced efficacy of climbing fibers to induce plasticity during VOR-decrease learning cannot be explained by a shorter complex spike duration.

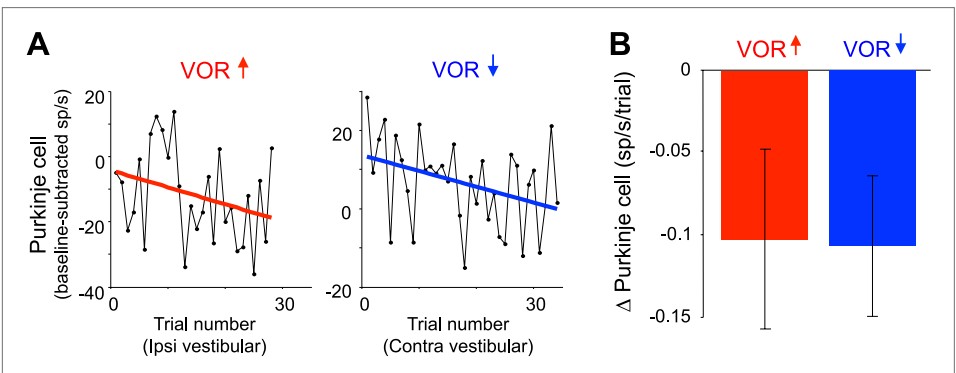

**Figure 4**. Changes in Purkinje cell output during the full VOR-increase and VOR-decrease training sessions. (**A**) Responses of example Purkinje cells during VOR-increase (red) and VOR-decrease (blue) training sessions, demonstrating a gradual change (reduction) in the firing rate of the two Purkinje cells over the course of ~30 individual trials. The baseline-subtracted simple spike firing rate was measured during the first 100 ms of each ipsiversive (VOR-increase) or contraversive (VOR-decrease) vestibular stimulus, and plotted as a function of trial number. The slope of the linear regression (colored lines) provided a measure of the rate of change in the Purkinje cell's response during the training session. (**B**) Average change in Purkinje cells during training, measured from the slopes of the linear regressions for all training sessions and all cells. There was a consistent reduction in Purkinje cell firing rate during contraversive vestibular stimuli over the course of ~90 s of VOR-decrease training (t(26) = -2.51, p=0.019, one sample t-test) and a trend for a decrease in firing during ipsiversive vestibular stimuli over the course of VOR-increase training (t(27) = −1.88, p=0.071, one sample t-test).

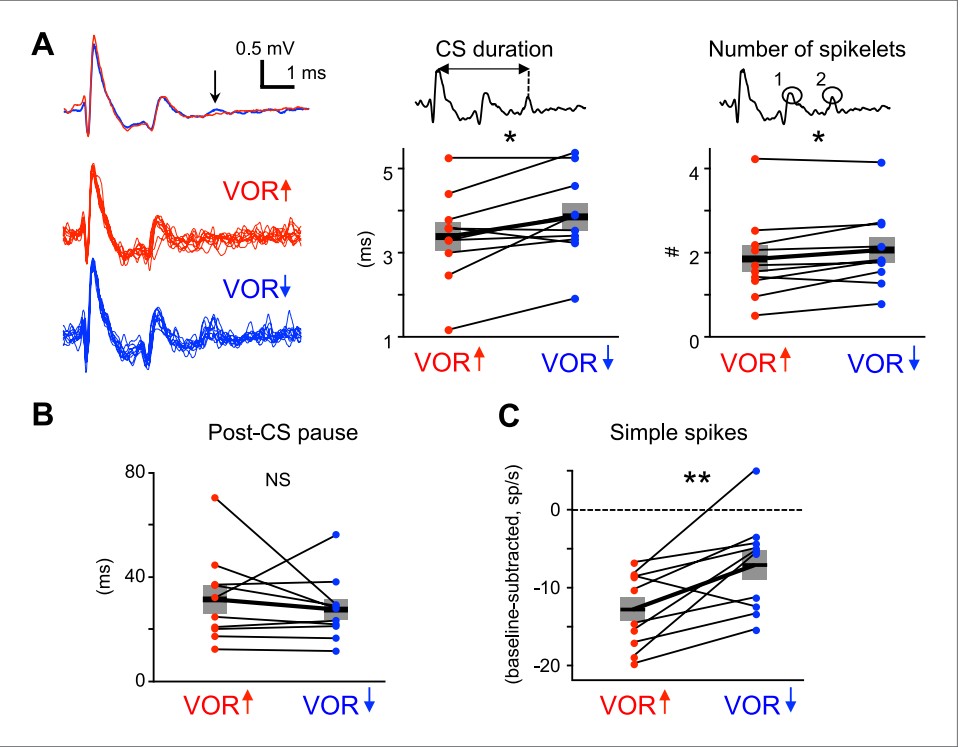

**Figure 5**. Measures of climbing fiber effects and circuit state during training. (**A**) Complex spike waveforms during VOR-increase (red) and VOR-decrease (blue) training. Left, example cell: top, mean waveforms; bottom, overlaid individual complex spikes. Note second spikelet during VOR-decrease training. Across all cells, the complex spike waveforms were of longer duration during VOR-decrease training (center, t(9) = −2.67, *p<0.05, paired *t*-test) and had more spikelets (right, t(9) = −2.71, *p<0.05, paired *t*-test) compared to VOR-increase training. Horizontal black bars and grey rectangles indicate mean ± SEM (**B**) The length of the pause in simple spike firing following a complex spike was similar during VOR-increase and VOR-decrease training (NS, t(9) = 0.79, p=0.45, paired t-test). Measurements in panels **A** and **B** were made on the same complex spikes used for the trial-by-trial analysis in *Figure 3*. (**C**) Mean Purkinje cell simple spike firing (baseline subtracted, first 250 ms of trials) during stimuli in the 'on' direction for the climbing fibers (*Figure 2E–H*) was higher during VOR-decrease training (blue), than during VOR-increase training (red; t(9) = −3.39, **p<0.01, paired t-test).

We also considered the possibility that the length of the climbing fiber-triggered pause in simple spike firing may affect the potency of climbing fiber activity to induce learning (*Maiz et al., 2012*). However, there was no significant difference in the duration of the post-complex spike pause during VOR-increase and VOR-decrease training (*Figure 5B*).

It is conceivable that climbing fiber-triggered plasticity would be less effective at reducing Purkinje cell simple spike firing if the firing rate was already very low. Therefore, we compared the average simple spike responses during vestibular stimuli in the direction associated with elevated climbing fiber activity: ipsiversive and contraversive vestibular stimuli during VOR-increase and VOR-decrease training, respectively. For both, firing rate decreased relative to baseline during the training stimulus (*Figure 2G,H*), but this reduction was smaller during VOR-decrease training than VOR-increase training (*Figure 5C*). This suggests that there is no floor effect limiting plasticity of the simple spike responses during VOR-decrease training. Nevertheless, the difference in simple spike rates during the two training paradigms suggests different patterns of excitatory and inhibitory input to Purkinje cells, which might regulate climbing fiber-triggered plasticity.

## Induction of learning with direct stimulation of the climbing fibers

The results of our trial-by-trial analysis suggest that error signals carried by climbing fibers contribute to the induction of plasticity during VOR-increase but not during VOR-decrease training. However, such correlational evidence cannot, by itself, establish causality, and the relationship between the

short-term effects observed in the trial-by-trial analysis and long-term learning has not been established (*Yang and Lisberger, 2013*). Thus, to provide a causal test of the climbing fiber role in VOR learning, and to analyze climbing fiber-induced plasticity over a longer time scale, we used an optogenetic stimulation approach in mice. We tested whether direct, optogenetic activation of climbing fibers could replace the sensory feedback (visual error signals provided by retinal slip) that normally drives VOR learning, and induce learning when paired with a vestibular stimulus in the absence of any visual feedback. We recently reported preliminary evidence that climbing fiber stimulation can induce VOR-increase but not VOR-decrease learning (*Nguyen-Vu et al., 2013*). However, the single set of stimulation parameters used in that study was designed to elicit a maximal response, and could potentially have recruited additional plasticity that masked the expression of any climbing fiber contribution to VOR-decrease learning. Moreover, there was no test of whether climbing fiber activation could be initiating changes in the VOR circuit that would support the delayed expression of VOR-decrease learning at later time points beyond the training session, which is plausible given the evidence that different mechanisms can support oculomotor learning over different time scales (*Titley et al., 2007*; *Shutoh et al., 2006*; *Okamoto et al., 2011*). Here, we address those limitations by testing a broader range of stimulation parameters and by measuring learning 2 hr after training as well as during the 30-min training session (*Figures 6 and 7*).

To optogenetically stimulate climbing fibers, virus carrying ChR2 was injected into the inferior olive, and several weeks later, the climbing fiber terminals in the cerebellar flocculus were illuminated with blue light (*Nguyen-Vu et al., 2013*). ChR2 expression in the climbing fibers was verified anatomically,

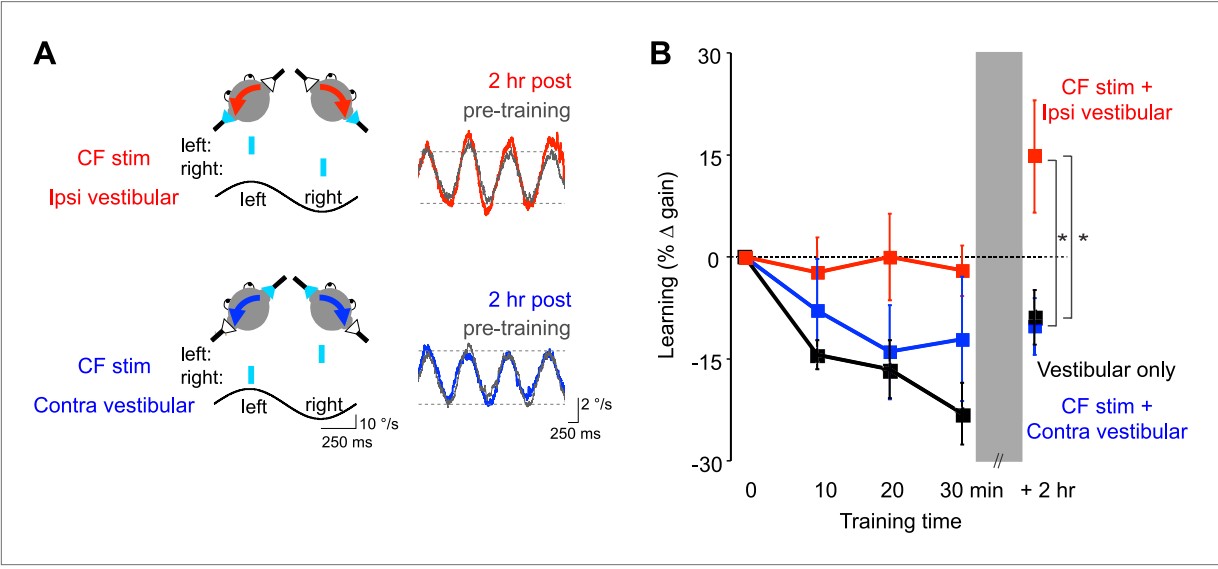

**Figure 6**. Optogenetic mimicry of error signals in the climbing fibers induced VOR-increase but not VOR-decrease learning. (**A**) Optogenetic training paradigms. Left, to mimic error signals carried by climbing fibers during visual-vestibular VOR-increase or VOR-decrease training (see *Figure 2A–F*), we optogenetically activated floccular climbing fibers during the ipsiversive phase (*CF stim + Ipsi Vestibular*, red) or contraversive phase (*CF stim + Contra Vestibular*, blue) of a 1 Hz sinusoidal vestibular stimulus (black). Climbing fibers were activated bilaterally using a single pulse of blue light repeated at 1 s intervals (cyan), see 'Materials and methods'). Right, before and after each training block, the VOR gain was measured in the absence of climbing fiber stimulation. Representative eye velocity traces from the same mouse pre-training (grey) and 2 hr after optogenetic VOR-increase training (red) or 2 hr after optogenetic VOR-decrease training (blue). (**B**) Motor learning induced by pairing climbing fiber activation with a vestibular stimulus. Learning was measured as the % change in VOR gain relative to pre-training, and depended on the training condition ($F_{(2,17)}$ = 3.81, p<0.05, repeated measures two-way ANOVA). When climbing fibers were activated during the ipsiversive phase of the vestibular stimulus (red), the VOR gain increased relative to control training with the vestibular stimulus in the absence of climbing fiber stimulation (*Vestibular-only*, black), and relative to training with climbing fiber activation paired with the contraversive phase of the vestibular stimulus (blue, *p<0.05, Fisher's post-hoc test). The changes in VOR gain induced by pairing climbing fiber activation with the contraversive vestibular stimulus were not significantly different from the vestibular-only control (p=0.62, Fisher's post-hoc test).

The following figure supplements are available for figure 6:

**Figure supplement 1**. Optogenetic activation of climbing fibers.

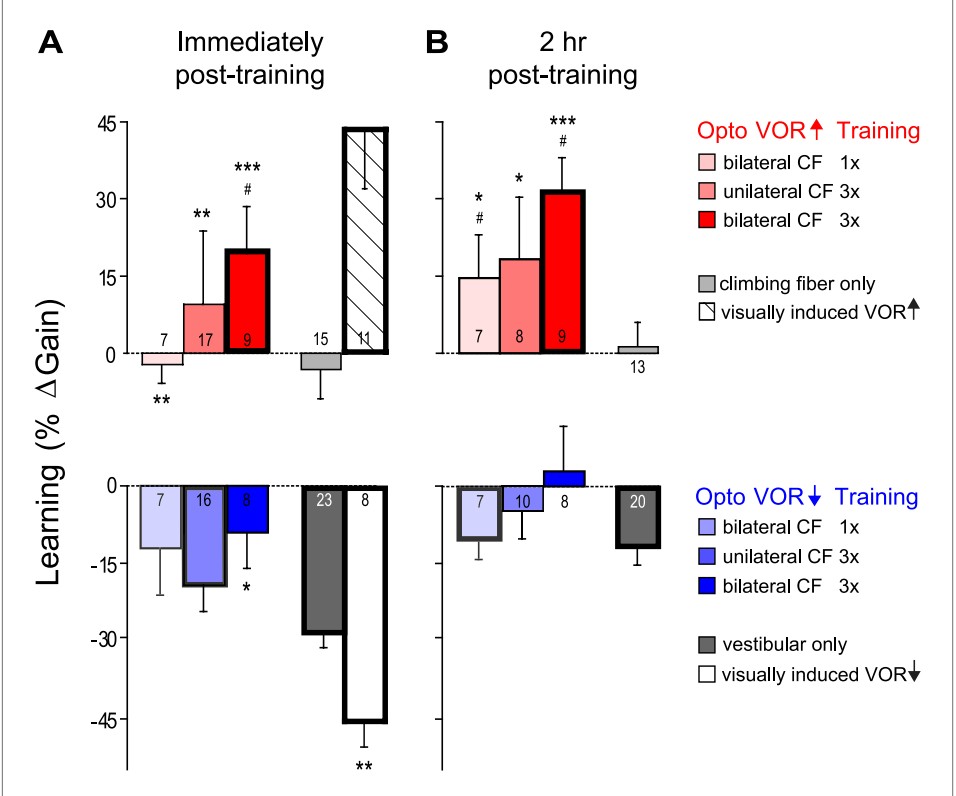

**Figure 7**. VOR learning induced by a range of climbing fiber stimulation protocols. VOR learning was measured immediately (**A**) and 2 hr after training (**B**). Climbing fibers were stimulated during the ipsiversive (red) or contraversive (blue) phase of the vestibular stimulus, to roughly mimic climbing fiber responses during visual-vestibular VOR-increase or VOR-decrease training, respectively. Climbing fibers were stimulated once (CF 1x) or three times (CF 3x) per cycle of the vestibular stimulus, unilaterally or bilaterally. Each bar represents the mean ± S.E.M change in VOR gain induced by each training paradigm relative to the pre-training baseline. Bars 'outlined in bold' are significantly different from zero (p<0.05); asterisks indicate significant difference from the vestibular-only control (*p<0.05; **p<0.01; ***p<0.001); # indicates the learned changes in VOR gain were different when the same climbing fiber stimulation was applied during the contraversive vs ipsiversive phase of the vestibular stimulus (red vs blue bars, p<0.05); one sample t-test, Wilcoxon signed rank test, or Mann–Whitney test (see 'Materials and methods'). Numbers indicate the number of mice for each training condition. Climbing fiber stimulation alone (light grey) induced no learning (p=0.12, Wilcoxon signed rank test for immediately post-training; p=0.79, one sample t-test for 2 hr post-training), but had a significant effect when paired with the ipsiversive phase of the vestibular stimulus (p<0.0001, Kruskal–Wallis test for training condition; post-hoc Dunn's multiple comparison tests vs vestibular-only, *p<0.05, **p<0.01, ***p<0.0001). Optogenetic VOR-decrease training with stimulation of the climbing fibers during the contraversive phase of the vestibular stimulus, to roughly mimic their response during VOR-decrease training (blue) did not induce an associative decrease in the VOR below the vestibular-only control (dark grey). Instead, there was a slight increase relative to vestibular-only immediately after training ($F_{(3, 50)}$ = 3.00, p<0.05, one-way ANOVA for training condition; *p<0.05, Dunnett's multiple comparison test), which was not significant 2 hr after training (p=0.49, Kruskal–Wallis test).

and optrode recordings from Purkinje cells demonstrated that complex spikes could be elicited by brief pulses of light to the flocculus (*Figure 6—figure supplement 1*).

To roughly mimic the pattern of climbing fiber activation observed during visually driven VOR-increase training (*Figure 2A,C,E*; *Ke et al., 2009*; *Raymond and Lisberger, 1998*; *Simpson and Alley, 1974*; *Watanabe, 1984*, *1985*), optogenetic climbing fiber activation was paired with ipsiversive vestibular stimuli. This pairing was done in total darkness, that is, in the absence of retinal slip, the sensory error signal that normally drives VOR learning. The vestibular stimulus consisted of 1 Hz sinusoidal head rotation. The climbing fiber stimulation was bilateral, with the climbing fibers in the flocculus on each side activated in alternation, using a single, 15-ms pulse of light centered on peak ipsiversive head

velocity: climbing fibers in the right flocculus were stimulated during peak rightward head velocity, and climbing fibers in the left flocculus were stimulated during peak leftward head velocity (*Figure 6A*, red). As a control, the same vestibular stimulus was delivered in the absence of any climbing fiber stimulation during the 30-min training period. This 'vestibular-only' training induced habituation of the VOR, a gradual decrease in VOR gain, as described previously by several groups (*Dow and Anastasio, 1999*; *Stahl, 2004*; *Boyden et al., 2006*; *Gutierrez-Castellanos et al., 2013*) (*Figure 6B*, black and *Figure 7A*, dark grey). If the climbing fibers were stimulated during the ipsiversive phase of the same vestibular stimulus, the VOR gain was higher than the vestibular-only control at the end of training (*Figure 6B*, red and *Figure 7A*, pink). This effect of climbing fiber stimulation was long-lasting; when retested 2 hr after training, the VOR gain was still higher compared to vestibular-only training (*Figure 6B*, red and *Figure 7B*, pink). For both training paradigms, there was an increase in VOR gain between the end of training and the 2 hr retest, presumably reflecting decay of the non-associative habituation (*Figure 6B*, black and *Figure 7*, dark grey).

To roughly mimic the pattern of climbing fiber activation during VOR-decrease training (*Figure 2B,D,F*) (*Ke et al., 2009*; *Raymond and Lisberger, 1998*; *Simpson and Alley, 1974*; *Watanabe, 1984*, *1985*), the climbing fibers were optogenetically activated during the *contraversive* phase of the vestibular stimulus (*Figure 6A*, blue). This pairing had no detectable effect, immediately or 2 hr post-training. There was no significant reduction of the gain of the VOR below the habituation level observed in response to training with the vestibular stimulus alone (*Figure 6B*, blue and *Figure 7*, light blue). In contrast, when the vestibular stimulus was paired with an appropriate visual stimulus (see 'Materials and methods'), there was a bigger decrease in VOR gain than induced by the vestibular stimulus alone, demonstrating that an associative decrease in VOR gain was possible (*Figure 7A*, white). Therefore, we tested whether stronger climbing fiber stimulation could induce an associative decrease in VOR gain.

We stimulated the climbing fibers three times during each cycle of the vestibular stimulus (125 ms inter-stimulus interval), either unilaterally or bilaterally. These stronger climbing fiber stimulation paradigms induced bigger increases in VOR gain when delivered during the ipsiversive phase of the vestibular stimulus (*Figure 7*, dark pink and red). However, when timed to induce VOR-decrease learning, increasing the number of climbing fiber stimuli had no effect (*Figure 7*, blue and dark blue).

Thus, none of our training paradigms induced a decrease in VOR gain below the vestibular-only control, either immediately or 2 hr after the end of training. It is possible that climbing fiber stimulation could have induced an associative decrease in VOR gain had we been able to more closely mimic the natural climbing fiber responses to a performance error. However, the same climbing fiber stimulation protocols were effective at inducing robust and graded learning when timed relative to the vestibular stimulus to mimic VOR-increase training. Moreover, the lack of VOR-decrease learning in response to optogenetic climbing fiber stimulation was consistent with the trial-by-trial analysis, which found no evidence for a contribution of the natural, visually-elicited error signals in the climbing fibers to the induction of VOR-decrease learning. Thus, the stimulation results, together with the trial-by-trial recordings, suggest that VOR-decrease learning is not induced by the climbing fibers, even though they carry error signals that could potentially guide learning, and the very same climbing fibers contribute to the induction of VOR-increase learning.

## Discussion

Both our recording and stimulation results indicate that during VOR learning, the efficacy of the climbing fibers for inducing plasticity is gated downstream of their encoding of errors. The unusually powerful synaptic connections between the climbing fibers and Purkinje cells have generally been viewed as providing a reliable and potent trigger for plasticity. In contrast, a comparison of the positive and negative results from the two VOR learning paradigms indicates that the coupling between climbing fiber activity and the induction of plasticity is regulated, and can vary across different learning contexts.

Our results provide some of the strongest evidence to date that error signals carried by the climbing fibers drive VOR-increase learning. However, the equally robust error signals carried by the climbing fibers during VOR-decrease training seem to make little or no contribution to the induction of learning. During VOR-decrease training, the animals are learning (*Raymond and Lisberger, 1996*), and there are changes in the responses of the Purkinje cells that can be detected within the 90-s training period (*Figure 4*). However, those changes do not seem to be driven by the climbing fibers, because there was no trial-by-trial correlation between climbing fiber activity and changes in Purkinje cell simple

spike responses of the kind observed during VOR-increase or smooth pursuit learning (*Figure 3*, *Medina and Lisberger, 2008*). Moreover, optogenetic stimulation of the climbing fibers at the time that they normally fire during VOR-decrease training did not induce VOR-decrease learning (*Figures 6 and 7*). Thus, in vivo, climbing fiber activity is sometimes a potent trigger for plasticity, but at other times has no effect. This gating of the climbing fibers' efficacy for inducing plasticity represents a novel component of the cerebellar learning algorithm.

## Convergent evidence from recording, stimulation and perturbation studies

The convergent evidence from recording and stimulation experiments, as well as previous, perturbation studies, strengthens the interpretation of the results. A previous stimulation study reported a failure of climbing fiber stimulation to induce VOR-decrease learning (*Nguyen-Vu et al., 2013*), however that negative finding was difficult to interpret, because it might have simply reflected a failure of the stimulation protocol used to adequately mimic the natural patterns of climbing fiber activation present during normal VOR-decrease learning. Therefore in the current paper, we paired stimulation experiments with highly complementary recording experiments, which showed that the *natural* climbing fiber responses present during visual-vestibular VOR-decrease training are not correlated with the induction of plasticity either. The stimulation experiments can demonstrate causality but are inherently 'unnatural'. The recording experiments document what happens under more natural conditions, but cannot establish causality. Together, the convergent evidence from the stimulation and recording experiments are considerably more powerful than either alone.

Moreover, the current recording and stimulation results are consistent with a previous perturbation study reporting that mice with impaired long-term depression of the parallel fiber-to-Purkinje cell synapses (pf-Pk LTD) are selectively impaired on VOR-increase but not VOR-decrease learning (*Boyden et al., 2006*). That study did not rule out a contribution of the climbing fibers to VOR-decrease learning via a mechanism other than pf-Pk LTD, such as rebound potentiation of inhibitory synapses onto the Purkinje cells (*Kano et al., 1992*; *Tanaka et al., 2013*) or plasticity in the Purkinje cells' targets caused by the climbing fiber-triggered pause in Purkinje cell simple spiking (*Maiz et al., 2012*). Thus, the present results extend our previous work by demonstrating a selective contribution, not only of pf-Pk LTD, but of all climbing fiber-dependent plasticity mechanisms to VOR-increase learning but not VOR-decrease learning. It is not known why the mechanisms for VOR-decrease and VOR-increase learning are different, although one can speculate that this would allow learning to appropriately weight the different 'costs' of having a VOR gain that is too high vs too low. For example, a VOR gain that is too high may be more likely to create eye movement instabilities.

The results for VOR-increase learning provide a positive control indicating that our experimental approaches were able to detect climbing fiber-triggered plasticity, which is critical for interpreting the negative results from VOR-decrease learning. Previous, seemingly inconsistent findings about the role of the climbing fibers in cerebellum-dependent learning have been difficult to reconcile because different labs use different behavioral paradigms and different experimental approaches. In contrast, direct comparison of the VOR-increase and VOR-decrease paradigms provides a demonstration that the very same animals and even the very same cells could toggle between either a tight coupling or no apparent coupling between climbing fiber activity and the induction of plasticity, depending on the behavioral context of the specific oculomotor training paradigm. Thus, direct comparison of the positive and negative results from VOR-increase and VOR-decrease learning in the same animals and the same cells, using the same techniques, suggests that climbing fiber efficacy is actively regulated.

## Cellular mechanisms gating climbing fiber efficacy

Recently, there has been considerable interest in the possibility that the duration of the complex spike may regulate the induction of plasticity by the climbing fibers. In vitro, ethanol and neuromodulators such as norepinephrine can affect the duration of a complex spike and the induction of climbing fiber-dependent plasticity (*Carey and Regehr 2009*; *He et al., 2013*; *Belmeguenai et al., 2008*). However, there was no reduction in complex spike duration during VOR-decrease training that would account for the reduced efficacy of climbing fiber spikes suggested by the trial-by-trial analysis.

Another factor that may gate the efficacy of climbing fibers to induce synaptic plasticity is the level of inhibition: coactive inhibitory inputs can impair the induction of LTD at the parallel fiber-to-Purkinje cell synapses (*Ekerot and Kano, 1985*), but also may serve as a substrate for climbing fiber-induced

plasticity (*Jörntell and Ekerot, 2003*; *Kano et al., 1992*; *Tanaka et al., 2013*). The level of inhibitory input is difficult to assess in vivo; however, we measured the rate of Purkinje cell simple spikes, which provides a readout of the net balance of excitation and inhibition. During both training paradigms, simple spike firing rate decreased relative to the spontaneous, pre-stimulus baseline around the time of climbing fiber activation (*Figure 5C*). This decrease in simple spike rate was smaller during VOR-decrease than VOR-increase training, indicating a higher ratio of excitatory to inhibitory inputs to the Purkinje cells at the time of climbing fiber activity during VOR-decrease vs VOR-increase training. Future studies will be required to determine whether this difference in E/I ratio could contribute to the differential efficacy of climbing fibers to trigger plasticity during the two training paradigms.

### Climbing fiber contribution to learning on different time scales

The relationship between short-term, trial-by-trial plasticity and longer-term learning is not understood (*Yang and Lisberger, 2013*). Our results provide some of the first evidence that the changes observed over different time scales may be mechanistically related, by showing parallels between the contribution, or lack of a contribution, of climbing fiber activity to plasticity on different time scales: a single trial, over the course of a 30-min training session, and 2 hr after the end of training.

In the trial-by-trial analysis for VOR-increase training, the change in Purkinje cell activity on trials following a climbing fiber spike was remarkably large, approximately 10% of the average firing rate (*Figure 3B*). If the 10% change observed in a single trial persisted in its entirety and accumulated across trials, there should be a 10,000% change during the ~1000 such trials that occur during an hour of training with the paradigms used in this study, but the observed change in behavior is typically less than 50% (*Pastor et al, 1992*; *Raymond and Lisberger, 1996*; *Boyden and Raymond, 2003*; *Katoh, 2007*; *Kimpo and Raymond, 2007*). This suggests that, at most, a small fraction (<1%) of the changes observed on a single trial could persist over the training session, consistent with a previous report that the single trial changes during smooth pursuit learning decay within a few seconds (*Yang and Lisberger, 2010*).

Nevertheless, the parallels between the changes observed on a single trial and the changes observed over longer time scales suggest that they could be related. In particular, the direction of the trial-by-trial changes in Purkinje cell responses during VOR-increase learning are consistent with those observed over the course of a brief, 90 s VOR-increase training period (*Figures 3B and 4*, red) and those observed after several hours or days of VOR-increase training (*Dufossé et al., 1978*; *Hirata and Highstein, 2001*; *Lisberger et al., 1994*; *Miles et al., 1980*; *Nagao, 1989*; *Watanabe, 1984*, *1985*). Moreover, our results from the trial-by-trial analysis were consistent with the changes we observed over tens of minutes to hours in the stimulation experiments (*Figures 6 and 7*), in that both suggested a contribution of the climbing fibers to the induction of VOR-increase but not VOR-decrease learning.

### Conclusion

Our results provide evidence for a novel component of the neural algorithm for cerebellar learning in vivo—namely, a gating of the ability of error signals in the climbing fibers to induce learning. Since previous work has shown that regulation can also occur at the level of olivary spiking, climbing fiber-triggered plasticity seems to be gated at two different stages—at the level of the olivary neurons responding to a performance error by either spiking or not spiking, and at the level of climbing fiber spikes either inducing or not inducing plasticity.

Thus, the cerebellum appears to conditionally heed the instructive signals provided by the climbing fibers. Dynamic gating of the neural error signals controlling the induction of learning may be a general feature of neural learning algorithms, and merits systematic investigation in the cerebellum and other circuits. Defining the factors that influence the receptivity of the cerebellum to neural error signals will be critical for developing more sophisticated theories of learning, and for developing strategies to optimize the recruitment of plasticity to aid learning or recovery from injury.

## Materials and methods

### Trial-by-Trial analysis of climbing fiber-associated plasticity
#### VOR training paradigms
During experiments, monkeys were restrained in a primate chair with their head fixed to the chair via a surgically implanted head post. Horizontal eye position (relative to the head) was measured using the

eye coil method (*Robinson, 1963*) and differentiated to obtain a record of eye velocity. Vestibular stimuli were delivered by rotating the primate chair and animal about an earth–vertical axis using a turntable controlled by a velocity servo motor (Carco Electronics). The vestibular stimulus was a 250-ms, 500-ms, or 1000-ms pulse of head velocity at 15°/s (25 ms of acceleration). The servo motor had a small overshoot at the end of the acceleration and deceleration (*Figure 2A,B*). Rightward and leftward vestibular stimuli were delivered in alternation, with a fixed interval of 2.192 s between vestibular stimuli in a given direction.

The vestibular stimuli were paired with a moving visual stimulus. The visual stimulus was provided by a high-contrast black-and-white pattern projected onto a tangent screen. Animals received a liquid reward for keeping their eyes within 2° of a small target in the middle of the visual stimulus. During VOR-increase training, the visual stimulus moved in the opposite direction from the vestibular stimulus at 15°/s (*Figure 2A*). During VOR-decrease training, the visual stimulus moved in the same direction as the vestibular stimulus at 15°/s (*Figure 2B*), so that the visual stimulus moved together with the head. These training stimuli induce a learned increase or decrease in VOR gain after 2–3 hr of training (*Raymond and Lisberger, 1996*). In the current experiments, we wanted to compare the responses of the same neurons to VOR-increase and VOR-decrease training stimuli. Therefore, the training stimuli were not presented for the prolonged training sessions that are typically used to induce learning, but rather the VOR-increase and VOR-decrease training stimuli were alternated in brief, 60–90 s blocks, so that the neural responses to the two kinds of training stimuli could be directly compared in the same neurons.

## Electrophysiology

Single-unit extracellular recordings were made from 10 Purkinje cells in the flocculus and ventral paraflocculus of two awake, behaving, male rhesus monkeys using platinum–iridium electrodes. Voltages related to neural activity were sampled at 50 kHz. The cells included in the analysis were all identified as horizontal gaze velocity Purkinje cells, based on their responses during smooth pursuit eye movements, VOR performance in the dark, and cancellation of the VOR as the monkey tracked a visual target moving exactly with the head (*Raymond and Lisberger, 1998*). Simple spikes were sorted using hardware window discriminators and digitized. Complex spikes were sorted off-line using Xwork (Lisberger Technologies) or Matlab (Mathworks, Inc., Natick, MA). Complex spikes provide a measure of the activity of the climbing fiber input to the Purkinje cell. Each Purkinje cell receives input from a single climbing fiber and each spike in that climbing fiber reliably triggers a complex spike in the Purkinje cell (*Eccles et al., 1966*).

## Data analysis

Voltages related to the eye velocity, vestibular stimulus, and visual stimulus were recorded at 500 Hz per channel. Eye velocity was obtained from the eye position signal using a hardware differentiator and filter with a 300 Hz corner frequency. For *Figure 2* and *Figure2—figure supplement 1*, retinal slip, the probability of a climbing fiber spike, and Purkinje cell simple spike rate were calculated from the responses to the 250-ms stimuli. Two cells were recorded only during 500 and 1000 ms training stimuli, and were thus excluded from the data shown in *Figure 2C–H*. However, for all quantitative analyses, all trials from all stimulus durations were included. To calculate the probability of a climbing fiber spike, data were binned in 50 ms bins, and the number of spikes within each bin was divided by the total number of trials. To calculate the retinal slip, saccades were first excised and replaced with a linear interpolation of the smooth eye velocity. Segments of data containing saccades that could not be interpolated were removed. Retinal slip (motion of the visual stimulus relative to the eye) was calculated by subtracting the sum of eye velocity (relative to the head) and head velocity (relative to the world) from the visual stimulus velocity (relative to the world). Average climbing fiber spike probability, retinal slip, and simple spike rate were calculated for each cell, and then combined to create a grand average. Simple spike firing rates were calculated using the reciprocal interval method (*Lisberger and Pavelko, 1986*). Since most Purkinje cells in the sample had some response to saccades, the firing rate was interpolated or excised according to whether the corresponding eye velocity trace was interpolated or excised. Simple spike firing rate was plotted in 10-ms bins in *Figure 2*. To account for slow changes in baseline firing rate, a 10 s moving average baseline was subtracted from simple spike rates in *Figures 3–5*. Error bars in all figures represent mean ± SEM.

Data were analyzed using Matlab to compute the changes in Purkinje cell firing on the trial following a trial with a climbing fiber spike. Trial-by-trial analysis was performed on consecutive vestibular

stimuli in the 'on' direction for climbing fibers: ipsilateral vestibular stimuli (head rotations towards the side of recording) for VOR-increase training and contraversive vestibular stimuli (away from the side of recording) for VOR-decrease training (*Figure 2A–F*). The intervening vestibular stimuli in the opposite direction were excluded from the analysis because of the very low rate of climbing fiber firing on those trials (*Figure 2—figure supplement 1A–F*). The analysis focused on the window 75–250 ms after stimulus onset, during which climbing fibers fired with high probability (*Figure 2E,F*, grey shading). Analyses were performed on pooled data from all training stimulus durations because the first 250 ms of these stimuli were identical.

For each training condition and each cell, we identified all pairs of consecutive trials in which a climbing fiber spike (CF) occurred in the analysis window on the first trial but not the second trial (No CF), and all pairs of consecutive trials in which there was no climbing fiber spike in the analysis window on either trial (No CF–No CF pairs). The simple spike response in the first trial was then subtracted from that in the second trial of the pair to obtain the trial-to-trial difference in the simple spike responses (*Figure 3A*). The trial-to-trial differences for all of the CF–No CF or No CF–No CF trial pairs recorded from an individual cell during a given training paradigm were then averaged separately, and convolved with a 25 ms sliding window. Finally, we averaged across cells to generate a grand average for each training condition. For this analysis, Purkinje cell simple spike rates were not analyzed beyond the first 100 ms of the trial (the approximate latency for climbing fiber responses) because of artifacts caused by the typical climbing fiber-triggered pause in simple spike firing.

To determine whether changes in Purkinje cell activity were significant, we created a bootstrap distribution of each data set and calculated the 95% confidence intervals (mean ± 2 standard deviation) based on this distribution. Mean values above or below the confidence intervals were considered significant. The bootstrap distribution was created as follows: for each cell, consecutive pairs of trials were selected randomly from the pool of all available trial pairs (including CF–No CF, No CF–No CF, No CF–CF, and CF–CF pairs) to match the number of CF–No CF or No CF–No CF pairs that were actually observed. An average trial-to-trial change in firing rate was calculated for each cell, and then all cells were averaged to complete one run of the bootstrap. The bootstrap was repeated 1000 times, and 95% confidence intervals were established based on the mean ± 2 standard deviation of the entire distribution.

To determine which statistical test is appropriate for each dataset in *Figure 4–7*, the normality of the distribution was first determined using the Shapiro–Wilk normality test (Graphpad Prism). Non-parametric tests were used when the distribution was not normal, and parametric tests when normal (Graphpad Prism or Matlab). All statistical analyses were performed as reported in the text. Significance level was set at 0.05.

To confirm that the Purkinje cells exhibited learning-related changes in their responses over the course of a VOR-decrease training session, even though there was no trial-by-trial effect of climbing fiber input during VOR-decrease training, we analyzed the progressive change in Purkinje cell firing over the course of each ~90-s training period. Previous studies have reported learning-related changes in the Purkinje cells' response during the VOR after at least an hour of VOR-increase or -decrease training (*Hirata and Highstein, 2001*; *Lisberger et al., 1994*; *Raymond and Lisberger, 1996*; *Watanabe, 1984*). To detect more subtle changes over rapid timescales, we plotted average Purkinje cell firing during the first 100 ms of each trial of a given training session as a function of trial number (typically, 30–40 individual vestibular stimuli, or trials, in a given direction made up a single session) (see examples in *Figure 4A*). Using linear regression, we fit a slope to this relationship, yielding a rate of change in the Purkinje cell's response, in units of spikes/s per trial, for each training session. To determine whether overall changes in Purkinje cell activity were significant, the rates of change from all sessions in each learning condition were compared to a null hypothesis of zero using a one sample t-test. We analyzed the first 100 ms of the trial because at that time the eye movement response is driven primarily by the VOR, with little influence of visually-driven eye movements.

For all the complex spikes used in the trial-by-trial analysis (i.e., within the analysis window of 75–250 ms, during contraversive vestibular stimuli for VOR-decrease training and ipsiversive vestibular stimuli for VOR-increase training), we measured features of the complex spike and the complex-spike-triggered pause in simple spiking. The number of spikelets following the large initial transient in each complex spike was counted manually, with the experimenter blind to training condition. The complex spike duration was defined as the latency from the peak of the initial large transient to the peak of the last spikelet. The length of the post-complex spike pause was calculated as the latency from each

complex spike to the next simple spike. Results were averaged within each cell, and a paired t-test between VOR-increase and VOR-decrease training sessions was performed to assess significance.

For the analysis shown in *Figure 5C*, average simple spike firing rates were measured during the first 250 ms of contraversive vestibular stimuli for VOR-decrease training and ipsiversive vestibular stimuli for VOR-increase training, and statistically compared using a paired t-test.

## Optogenetic stimulation of the climbing fiber to induce learning

An optogenetic approach was used to stimulate the climbing fibers (*Boyden et al., 2005*). The neurons in the inferior olive whose axons form the climbing fibers in the left and right cerebellar flocculus are in the right and left dorsal cap of Kooy, respectively, which lie on either side of the midline (*Ruigrok et al., 1992*; *Tan et al., 1995*; *Sugihara and Shinoda, 2004*). Retinal slip in a given direction drives opposite responses (increases vs decreases in firing) in dorsal cap neurons on opposite sides of the midline. Therefore, to approximate the natural responses of the climbing fibers to retinal slip requires independent control of olivary neurons on either side of the midline, which would be difficult to accomplish using electrical stimulation. The optogenetic approach overcomes this limitation, because ChR2-expressing climbing fibers can be selectively activated on one side of the brain by shining light on their terminals in the cerebellar flocculus.

### Surgeries

Experiments were performed on adult C57BL/6 male mice, as previously described (*Nguyen-Vu et al., 2013*). To express ChR2 in the climbing fibers of the flocculus on both sides of the cerebellum, we injected AAV-CamKIIα-ChR2(H134R)-eYFP (Karl Deisseroth, Stanford University or Neuroscience Gene Vector and Viral Core, Stanford University; titer $\geq 10^{12}$) into the dorsal cap of Kooy of the inferior olive, which provides the climbing fiber input to the cerebellar flocculi, of mice $\geq$9 weeks old. Stereotaxic coordinates were used to target the dorsal cap (*Nishiyama and Linden, 2004*; *Grasselli et al., 2011*). A 0.5–1.0 µl volume of viral particles was injected over the course of 15–30 min. Mice were surgically prepared for behavioral experiments (*Boyden and Raymond, 2003*) 6–8 weeks after the virus injection. Briefly, a headpost was attached to the skull using anchor screws and dental acrylic. An eye coil (IET, Marly, Switzerland) was implanted beneath the conjunctiva in one eye, so that horizontal eye position could be tracked using the eye coil method. To provide access to the cerebellar flocculi for optical stimulation and electrophysiology, a craniotomy was performed on the periotic capsule on each side of the head, and a cannula was implanted over each craniotomy using dental acrylic. After surgery, mice were allowed 5–7 days to recover before behavioral experiments were conducted.

### Optrode recordings

To verify that optogenetic stimulation could effectively drive spikes in the relevant climbing fibers, in vivo extracellular recordings of Purkinje cells in the cerebellar flocculus were performed using an 'optrode', a tungsten electrode (FHC) coupled to a 200-µm optical fiber (Thor Labs; n = 5 cells in four mice). The optrode was introduced via the implanted cannula. For some physiological recordings, the mouse was initially anesthetized using isoflurane or ketamine and then allowed to recover for the recordings. Purkinje cell activity was sampled at 50 KHz using Spike 2 (CED), and sorted off-line (Spike 2). Climbing fibers were activated using the same optical stimulation parameters that were used in the behavioral, optogenetic training experiments (see below). An electrophysiological waveform artifact due to the light pulse was observed in some recordings. To isolate the waveform artifact, the optrode was moved until the activity of the cell that was being recorded disappeared, and the same light stimulus trains were then delivered. This isolated waveform of the artifact was subtracted from the electrophysiological traces to identify the complex spike response to the light pulse.

### Behavioral experiments

During training, the mouse's head was immobilized by attaching the implanted headpost to a restrainer, which was mounted on a servo-controlled turntable (Carco Electronics). Vestibular stimuli were delivered by rotating the whole body about an earth–vertical axis using a 1 Hz sinusoidal velocity profile (±10°/s peak velocity). To stimulate the climbing fibers in the flocculus, a 200-µm optical fiber (Thor Labs) was introduced through the cannula implanted above the cerebellar flocculus. To optimize the position of the optical fiber for the most effective activation of the climbing fibers, a 20-Hz train of 2 ms light pulses was delivered for 3 s at different depths. When positioned near climbing fibers expressing ChR2, the high frequency stimulation induced a nystagmus similar to what was previously

described when the inferior olive was electrically stimulated (**Barmack and Hess, 1980**). The optical fiber was placed at the depth at which the largest eye movements were induced. After several minutes of 'rest' time, a single light pulse, or a 250-ms train of three light pulses with a 125 ms inter-pulse-interval (473 nm light, 15 ms pulse duration, 1–2 mW/mm$^2$) was repeated every second and centered on peak ipsiversive or contraversive vestibular stimulus speed, to roughly mimic the climbing fiber responses during VOR-increase or VOR-decrease training, respectively. Stimulation was delivered bilaterally, with stimulation of climbing fibers in the two cerebellar hemispheres 180° out of phase. In a separate cohort of mice, a 250-ms train of three light pulses with a 125-ms inter-pulse-interval (2, 10 or 15 ms pulse duration, 0.3–1 mW/mm$^2$) was delivered unilaterally. As a control, the same experiments were performed without climbing fiber stimulation during the training blocks ('vestibular only' condition). In a 'climbing fiber only' training condition, climbing fibers were stimulated bilaterally or unilaterally with the train of three light pulses used in optogenetic training. The effects of unilateral or bilateral climbing fiber only stimulation were not significantly different, and were therefore pooled (p=0.21, unpaired *t*-test; **Figure 7**, light grey).

Training periods consisted of 3 × 10-min blocks of vestibular stimuli paired with climbing fiber stimulation. Before and after each block, the vestibular stimulus was delivered in the absence of climbing fiber stimulation for 60–80 s to measure the gain of the VOR. The VOR was measured again 2 hr after optogenetic training; during the intervening two-hour retention period, the animal was kept in the dark in its home cage. All training and testing were conducted in total darkness; the optical fiber used to optically stimulate the flocculus was optically insulated to prevent light leaks.

For each mouse, the order of training with the optogenetic VOR-increase, optogenetic VOR-decrease, vestibular only and climbing fiber only conditions was pseudo-randomized. For the mice that underwent bilateral climbing fiber stimulation, half were tested first with the single light pulse simulation protocol, and the other half with the train of three light pulses. There were at least 48 hr between tests of the different training conditions. After the optogenetic training experiments were conducted, we tested the ability of the mice to undergo a learned decrease and increase in VOR gain in response to a more typical VOR training paradigm, which paired the vestibular stimulus with a visual stimulus. For visually-induced VOR-decrease or VOR-increase training, the vestibular stimulus (1 Hz, ±10°/s peak velocity) was paired with a visual stimulus that moved exactly with or exactly opposite the head, respectively. The visual stimulus was provided by a back-lit optokinetic drum with black and white stripes, each subtending approximately 7.5° of visual angle.

For bilateral stimulation with single light pulses, 7 mice were tested on all training conditions, except one mouse was not tested in the vestibular only condition. For bilateral stimulation with three light pulses, 7 mice were tested on all training conditions, and an additional two mice were tested on a subset of training conditions. Data from the 30 min time point of the bilateral experiments with three light pulses, along with the corresponding vestibular only controls from the same mice, were reported in a previous study (**Nguyen-Vu et al., 2013**). For unilateral stimulation experiments, eight mice were tested on all training conditions, and an additional 16 mice were tested on a subset of training conditions. Eight mice from the cohort used for unilateral stimulation were also tested on visual-vestibular VOR-decrease training.

## Data analysis

Eye position, vestibular and visual stimulus position and velocity were sampled at 1000 Hz (Spike 2). Custom software was used for data analysis (Mathworks, Inc.). To calculate eye velocity, the eye position was first low-pass filtered using a third-order Butterworth filter with a cutoff frequency of 9 Hz, and then differentiated. Eye velocity data were edited to remove saccades and any body movement artifacts using a velocity threshold. Eye and head velocity data were fit with sine waves. VOR gain was calculated by dividing the amplitude of the eye velocity fit by the amplitude of the head velocity fit. Learning was calculated as the percent change in VOR gain relative to the pre-training baseline, that is, a 0% change would indicate no learning. The normality of the distribution of VOR gain changes for each training condition at each time point of interest was tested using the Shapiro–Wilk normality test (Graphpad Prism). For statistical analyses that included at least one set of data with a non-normal distribution, non-parametric tests were used. For the experiments that used single pulses of climbing fiber stimulation (**Figure 6B**), a repeated measures two-way ANOVA, with training condition and time as factors, was performed (Statview) to test whether climbing fiber stimulation induced changes in VOR gain. To determine which training conditions were significantly different from each other, a

Fisher's post-hoc test was performed. To test whether a change in VOR gain was significantly different from pre-training baseline (i.e., from zero) immediately or 2 hr post-training, a one sample t-test (parametric) or a Wilcoxon Signed Rank Test (non-parametric) was performed for each training condition (*Figure 7*). To test whether training paradigms that used different amounts of climbing fiber stimulation had different effects, immediately or 2 hr after training (*Figure 7*), a one-way ANOVA (parametric) or Kruskal–Wallis test (non-parametric) was performed, with training condition (bilateral CF 3x, unilateral CF 3x, bilateral CF 1x and vestibular only) as the factor (see *Figure 7*). When appropriate, a Dunnett's (parametric) or Dunn's (non-parametric) multiple comparisons test was performed to compare each optogenetic training paradigm to the vestibular only control (Graphpad Prism). This analysis was conducted separately for the pairing of optogenetic stimulation with an ipsiversive vs contraversive vestibular stimulus. To test whether the visually induced learned decreases in the gain of the VOR were different from the changes induced by the vestibular only training, an unpaired t-test was performed (*Figure 7*). To compare the effects of stimulating the climbing fibers during the ipsiversive vs contraversive phase of the vestibular stimulus, an unpaired t-test (parametric) or Mann–Whitney test (non-parametric) was performed on the changes in VOR gain measured immediately or 2 hr post-training (*Figure 7*). Significance level was set at 0.05.

All procedures were approved by Stanford University's Administrative Panel on Laboratory Animal Care.

## Acknowledgements

We thank S Lisberger and Y Yang for helpful discussions; M Goldman, V McGinty and members of the Raymond Lab for helpful comments on earlier versions of the manuscript; TDB Nguyen-Vu, G Zhao for expert technical advice and R Hemmati and S Umamoto for technical assistance.

This work was supported by the US National Institutes of Health grant K01 NS069617 (RRK), supplement to P01 NS053862 (RRK), R01 NS072406 (JLR), 5T32MH020016-14 (CKK), the Weston Havens Foundation (JLR), the National Science Foundation GFRP DGE-114747 (HLP) and the IGERT Fellowship 0801700 from the Stanford Center for Mind, Brain, and Computation (JMR and HLP).

## Additional information

### Funding

| Funder | Grant reference number | Author |
| --- | --- | --- |
| US National Institutes of Health | R01 NS072406 | Jennifer L Raymond |
| US National Institutes of Health | R01 DC04154 | Jennifer L Raymond |
| The Weston Havens Foundation | | Jennifer L Raymond |
| US National Institutes of Health | K01 NS069617 | Rhea R Kimpo |
| US National Institutes of Health | T32 MH020016 | Christina K Kim |
| US National Science Foundation, Fellowsip from Stanford Mind Brain Computation IGERT | 0801700 | Jacob M Rinaldi, Hannah L Payne |
| US National Science Foundation, Graduate Research Fellowship | DGE-114747 | Hannah L Payne |

The funders had no role in study design, data collection and interpretation, or the decision to submit the work for publication.

### Author contributions

RRK, JMR, JLR, Conception and design, Acquisition of data, Analysis and interpretation of data, Drafting or revising the article; CKK, HLP, Conception and design, Analysis and interpretation of data, Drafting or revising the article

### Ethics

Animal experimentation: This study was performed in accordance with the recommendations in the National Institutes of Health's Guide for the Care and Use of Laboratory Animals. All procedures were approved by Stanford University's Administrative Panel on Laboratory Animal Care (Protocol #9143,

10907). All surgery was performed under isoflurane anesthesia and every effort was made to minimize suffering.

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
