## [Decision Letter]

Thank you for sending your work entitled “Gating of neural error signals during motor learning” for consideration at *eLife*. Your article has been evaluated by a Senior editor, a Reviewing editor, and 2 reviewers.

The Reviewing editor and the reviewers discussed their comments before we reached this decision, and the Reviewing editor has assembled the following comments to help you prepare a revised submission. This invitation to submit a revised manuscript is not a guarantee of eventual acceptance.

This could be a potentially very important manuscript, but reviewers felt that the important message does not come across. Because the paper deals with a number of different topics, it is sometimes hard to follow and the innovation/conceptual advances often remain unclear. Several observations are interesting, but as written it is not obvious that the paper provides a cohesive story and a major conceptual advance. Thus, the authors can and should boost up both Introduction and Discussion to highlight what is unique and novel in the present experiments. Also the Results section needs to provide better logical justification about the numerous experiments performed. The potential importance of the paper was better articulated during the consultation process and discussion among the reviewers and editors than by the manuscript itself. The case for the conceptual value of the manuscript must be made by the paper itself and the new data it presents.

The biggest concern with the manuscript is that the progress is several small loosely linked steps forward and there was concern that the major advances are far from clear. There was overall concern about the fact that stimulation experiments are in mice and recording experiments in monkey. The relationship between trial-by-trial learning and longer-term learning is not established and although reviewers thought this one of the most interesting aspects of the paper, the connection to the rest of the paper was not clear. One of the reviewers expressed serious reservations about these shortcomings. It is important that more focus on the novel aspects of the paper is needed. In addition, the authors should either eliminate some of the less compelling aspects of the paper or fill in the dots as to why they are important and related to the big story. Some more specific comments are provided next.

1) The paper deals with multiple topics that are not well integrated to form a cohesive story. In many ways the study of trial-by-trial of Figure 3 is reminiscent of the recent Lisberger *eLife* paper that provided an in-depth characterization of trial-by-trial learning in smooth pursuit. In contrast to that paper, the present manuscript does not provide an in depth study of trial-by-trial learning. Although there is speculation about the relationship between trial-by-trial learning and long-term learning, it is not known if these phenomena are related in any way. A clear distinction and comparison is necessary. The differences between pursuit versus VOR should also be discussed.

2) Previous studies showed that VOR-increase and VOR-decrease learning are mediated by different mechanisms (8; 7). It was already known that there could be climbing fiber activity that would drive VOR-increase learning but not VOR decrease learning. As a result, the findings of Figure 5 do not appear particularly surprising, so the authors have to motivate the novelty better.

3) The ChR2 experiments are interesting but they are very closely related to the authors' recent Nature Neuroscience paper describing similar results. It is not clear whether the data in the current study provide a significant advance over [64]: They use some of the same control data in Figures 6 and 7, and in that paper they showed that optogenetic activation could induce VOR-increase learning but not VOR-decrease learning.

4) The term gating might be somewhat confusing in the context of this paper because it has previously been used to refer to gating in the inferior olive (Andersson & Armstrong, Apps et al. Hesslow et al. Gibson et al.), for instance because of the nucleo-olivary pathway. When using the term “gating” (as in gating in the olive or gating pain signals) one would normally suggest that it is a design principle of a circuit that a signal with a particular function is sometimes allowed, sometimes not, to fulfil that function. When A causes B only when it occurs together with C, we do not usually say that C is a gating mechanism. In the reviewers' view, the authors seem to suggest something stronger than what they have demonstrated. When it comes to the speculations about what the missing factor might be, I think that a word or two about how the mossy fiber input might differ between the two experimental situations might be justified.

5) The conclusion that the ability of climbing fibers to induce cerebellar plasticity is regulated downstream, rather than in a different area, should be better justified.

6) It would also be nice to discuss why increases and decreases in VOR gain might be implemented by different mechanisms. The reviewers appreciate that such discussion might be hypothetical at present, yet it is important to be included in the paper.

---

## [Author Response]

*This could be a potentially very important manuscript, but reviewers felt that the important message does not come across. Because the paper deals with a number of different topics, it is sometimes hard to follow and the innovation/conceptual advances often remain unclear. Several observations are interesting, but as written it is not obvious that the paper provides a cohesive story and a major conceptual advance. Thus, the authors can and should boost up both Introduction and Discussion to highlight what is unique and novel in the present experiments. Also the Results section needs to provide better logical justification about the numerous experiments performed. The potential importance of the paper was better articulated during the consultation process and discussion among the reviewers and editors than by the manuscript itself. The case for the conceptual value of the manuscript must be made by the paper itself and the new data it presents*.

*The biggest concern with the manuscript is that the progress is several small loosely linked steps forward and there was concern that the major advances are far from clear*.

The manuscript has been extensively rewritten to clarify the rationale for all of the experiments and to more effectively highlight the novel findings and conclusions.

The major conceptual advance is the demonstration of a new component of the learning algorithm in the cerebellum of awake behaving animals, namely regulation of the ability of climbing fiber spiking to trigger cerebellar plasticity. The Marr-Albus-Ito model of cerebellar learning suggests that a performance error elicits a response in the climbing fibers, which triggers plasticity. Previous studies have provided evidence that the ability of error-related cues to drive spiking in the climbing fibers can be regulated by the state of the cerebellar circuit during training ([100]; [72]; [27]; [3]; reviewed in Apps et al. 1999 and ; [20]). Our paper shows another level of regulation – even when there is a robust response in the climbing fibers, its ability to trigger plasticity can also be regulated by the state of the cerebellar circuit. Related phenomena have been described in vitro (11; 55); our paper is the first to show that this regulation can occur in intact animals undergoing learning. Thus, the cerebellum is not a slave to its climbing fiber ‘teachers,’ but plays an active role in determining whether it will adapt in response to the error signals it receives from the climbing fibers, with multiple stages of regulation.

*There was overall concern about the fact that stimulation experiments are in mice and recording experiments in monkey*.

The VOR is a phylogenetically old behavior, and we know of no evidence to suggest mechanistic differences across species. On the contrary, decades of results reported from fish, rodents, rabbits, cats, monkeys and humans have been remarkably consistent, as is the case for the present experiments.

The recordings and associated trial-by-trial analysis of VOR learning were conducted in rhesus monkeys so that they could be directly compared with the in-depth, trial-by-trial analysis of monkey smooth pursuit learning described in a recent *eLife* paper from the Lisberger lab and two previous papers from that lab. Those papers reported that a single complex spike in a Purkinje cell, reflecting a spike in its climbing fiber input, reliably triggers plasticity that can be detected on the next trial during smooth pursuit learning. In striking contrast to those papers, we found a clear dissociation between the complex spikes and the induction of plasticity during VOR-decrease learning. Because the experiments were all done in rhesus monkeys, this difference, which is central to our conclusions, cannot be attributed to a species difference.

The optogenetic stimulation experiments were conducted in mice for practical reasons; the technical difficulty of doing those experiments in monkeys was prohibitive.

*The relationship between trial-by-trial learning and longer-term learning is not established and although reviewers thought this one of the most interesting aspects of the paper, the connection to the rest of the paper was not clear*.

The relationship between trial-by-trial learning and longer term learning is not understood, but our results provide some of the best evidence to date that they may be related, by showing parallels between the contribution, or lack of a contribution, of climbing fiber activity to plasticity on several different time scales. The contribution of the climbing fibers to plasticity differed across learning paradigms, however, within a paradigm, the contribution of the climbing fibers was consistent over a single trial, over the course of a 30-minute training session, and 2 hours after the end of training.

*One of the reviewers expressed serious reservations about these shortcomings. It is important that more focus on the novel aspects of the paper is needed. In addition, the authors should either eliminate some of the less compelling aspects of the paper or fill in the dots as to why they are important and related to the big story*.

We have extensively rewritten the manuscript to focus on the main, novel conclusions and to weave them more tightly together.

*1) The paper deals with multiple topics that are not well integrated to form a cohesive story. In many ways the study of trial-by-trial of*
Figure 3
*is reminiscent of the recent Lisberger eLife paper that provided an in-depth characterization of trial-by-trial learning in smooth pursuit. In contrast to that paper, the present manuscript does not provide an in depth study of trial-by-trial learning. Although there is speculation about the relationship between trial-by-trial learning and long-term learning, it is not known if these phenomena are related in any way. A clear distinction and comparison is necessary. The differences between pursuit versus VOR should also be discussed*.

The trial-by-trial analysis technique used in our manuscript was modeled on that used by the Lisberger lab, moreover, we are recording from Purkinje cells in the same region of the cerebellum, the floccular complex. Despite the similarities in our experimental approaches, our results and conclusions differ in a major way from the Lisberger lab’s results on smooth pursuit learning. The previous analyses of smooth pursuit learning (and our analysis of VOR-increase learning) all found a consistent, trial-by-trial correlation between spikes in a climbing fiber and the induction of plasticity, consistent with the classic, Marr-Albus-Ito model of cerebellar learning. In striking contrast, our results from VOR-decrease learning show that there are conditions where there are robust error signals and spiking in the climbing fibers, but there is no induction of plasticity that can be detected on the next trial. This latter observation supports our novel conclusion that the ability of the climbing fibers to induce plasticity can be gated downstream of spikes in the climbing fibers. The revised manuscript lays out this comparison of the different oculomotor learning tasks. In addition, we include a discussion of the different time scales of plasticity.

*2) Previous studies showed that VOR-increase and VOR-decrease learning are mediated by different mechanisms (*[8]*;*
[7]*). It was already known that there could be climbing fiber activity that would drive VOR-increase learning but not VOR decrease learning. As a result, the findings of*
Figure 5
*do not appear particularly surprising, so the authors have to motivate the novelty better*.

The revised manuscript better delineates the novelty of the current findings relative to previous work from our laboratory. Boyden & Raymond 2004 hypothesized that different mechanisms contribute to VOR-increase and VOR-decrease learning, based purely on behavioral observations of the different properties of these two forms of VOR learning. [7] provided evidence that one specific climbing fiber-triggered plasticity mechanism, long-term depression of the parallel fiber-to-Purkinje cell synapses, (pf-Pk LTD) contributes selectively to VOR-increase but not VOR-decrease learning. That study left open the possibility that the robust error signals carried by the climbing fibers during VOR-decrease training might contribute to VOR-decrease learning through a mechanism other than pf-Pk LTD, such as rebound potentiation of the inhibitory synapses onto the Purkinje cells ([35]; Tanaka et al. 2013) or plasticity in the vestibular nuclei induced by the climbing fiber-triggered pause in simple spiking (52). Our current manuscript assesses the contribution, not only of pf-Pk LTD, but the net effect of all climbing fiber-dependent plasticity mechanisms, and finds a selective contribution to VOR- increase learning, but not VOR-decrease learning, despite the equally robust responses in the climbing fibers during both paradigms.

*3) The ChR2 experiments are interesting but they are very closely related to the authors' recent Nature Neuroscience paper describing similar results. It is not clear whether the data in the current study provide a significant advance over*
[64]*: They use some of the same control data in*
Figures 6 and 7*, and in that paper they showed that optogenetic activation could induce VOR-increase learning but not VOR-decrease learning*.

The main conclusion of [64] was that Purkinje cell simple spike activity can drive learning. In order to demonstrate that Purkinje cell simple spike activity triggers plasticity via a mechanism distinct from that triggered by climbing fiber activity, we also included in that paper a few control experiments using optogenetic stimulation of the climbing fibers. The conclusion we drew from those experiments was that climbing fiber activation and Purkinje cell activation could not substitute for each other. Those experiments provided a hint that climbing fiber activation does not drive VOR-decrease learning; however, the data were not sufficient to make that conclusion. Most importantly, the climbing fiber stimulation might have failed to induce VOR-decrease learning simply because it failed to adequately mimic the natural patterns of climbing fiber activation present during normal VOR-decrease learning. Therefore in the current manuscript, we pair the stimulation experiments with highly complementary recording experiments, which show that the *natural* climbing fiber responses present during visual-vestibular VOR- decrease training are not correlated with the induction of plasticity either. Each approach has its strengths and limitations – the stimulation experiments can establish causality but are inherently “unnatural”, whereas the recording experiments document what happens under more natural conditions, but can only show correlations, which does not necessarily mean there is a causal relationship. Together the convergent evidence from the stimulation and recording experiments are considerably more powerful than either alone, and thus the combination of the two is an important contribution of the current manuscript. In addition, the current manuscript improves upon the stimulation approach by comparing multiple climbing fiber stimulation protocols and by testing whether climbing fiber stimulation might be initiating changes in the VOR circuit that would support the delayed expression of VOR-decrease learning, which does not appear to be the case.

*4) The term gating might be somewhat confusing in the context of this paper because it has previously been used to refer to gating in the inferior olive (Andersson & Armstrong, Apps et al. Hesslow et al. Gibson et al.), for instance because of the nucleo-olivary pathway. When using the term “gating” (as in gating in the olive or gating pain signals) one would normally suggest that it is a design principle of a circuit that a signal with a particular function is sometimes allowed, sometimes not, to fulfil that function. When A causes B only when it occurs together with C, we do not usually say that C is a gating mechanism. In the reviewers' view, the authors seem to suggest something stronger than what they have demonstrated. When it comes to the speculations about what the missing factor might be, I think that a word or two about how the mossy fiber input might differ between the two experimental situations might be justified*.

We use the term “gating” to describe our observation that in the very same animals and the very same cells, there can either be a tight coupling between climbing fiber spikes and the induction of plasticity, or no such coupling, depending on the behavioral context. The revised Discussion clarifies that climbing fiber triggered plasticity seems to be regulated, or gated, at two different stages – at the level of the olivary neurons either spiking or not spiking, for instance because of the Purkinje cell-nucleo-olivary pathway; and at the level of spikes in the olivary neurons and their climbing fiber axons either inducing or not inducing plasticity.

We now include a discussion of what is known about the mossy fiber input in the two relevant contexts, which suggests that some other factor provides the gate.

*5) The conclusion that the ability of climbing fibers to induce cerebellar plasticity is regulated downstream, rather than in a different area, should be better justified*.

Two lines of evidence suggest that we are recording from and stimulating the cerebellar area (the floccular complex) relevant for both learning paradigms. First, lesion studies have shown deficits in both VOR-increase and VOR-decrease learning after lesions of the floccular complex ([78]; [31], Lisberger et al. 1984; [62], Rambold et al. 2002; [57]). Second, we could detect plasticity in the simple spike responses of the Purkinje cells we recorded over the course of the 90-sec training period during VOR-decrease as well as VOR-increase training (Figure 4). This suggests that we were recording not only in the relevant area of the cerebellum, but also in the relevant cells for both VOR-increase and VOR-decrease learning.

*6) It would also be nice to discuss why increases and decreases in VOR gain might be implemented by different mechanisms. The reviewers appreciate that such discussion might be hypothetical at present, yet it is important to be included in the paper*.

We now include a brief treatment of this issue in the Discussion.